# Amine-Functionalized Natural Halloysite Nanotubes Supported Metallic (Pd, Au, Ag) Nanoparticles and Their Catalytic Performance for Dehydrogenation of Formic Acid

**DOI:** 10.3390/nano12142414

**Published:** 2022-07-14

**Authors:** Limin Song, Kaiyuan Tan, Yingyue Ye, Baolin Zhu, Shoumin Zhang, Weiping Huang

**Affiliations:** 1College of Chemistry, Nankai University, Tianjin 300071, China; 17853135525@163.com (L.S.); tky8246159357@163.com (K.T.); aniy1110@hotmail.com (Y.Y.); zhangsm@nankai.edu.cn (S.Z.); 2The Key Laboratory of Advanced Energy Materials Chemistry (Ministry of Education), Nankai University, Tianjin 300071, China

**Keywords:** halloysite nanotube, formic acid, dehydrogenation, γ-aminopropyltriethoxysilane, Pd

## Abstract

In today’s age of resource scarcity, the low-cost development and utilization of renewable energy, e.g., hydrogen energy, have attracted much attention in the world. In this work, cheap natural halloysite nanotubes (HNTs) were modified with γ-aminopropyltriethoxysilane (APTES), and the functionalized HNTs were used as to support metal (Pd, Au, Ag) catalysts for dehydrogenation of formic acid (DFA). The supports and fabricated catalysts were characterized with ICP, FT-IR, XRD, XPS and TEM. The functional groups facilitate the anchoring of metal particles to the supports, which brings about the high dispersion of metallic particles in catalysts. The catalysts show high activity against DFA and exhibit selectivity of 100% toward H_2_ at room temperature or less. The interactions between active centers and supports were investigated by evaluation and comparison of the catalytic performances of Pd/NH_2_-HNTs, PdAg/NH_2_-HNTs and PdAu/NH_2_-HNTs for DFA.

## 1. Introduction

Under the general trend of global energy conservation and emission reduction, replacing fossil energy with clean and renewable energy is getting more and more attention [1,2,3,4]. Hydrogen (H_2_), an important renewable energy carrier, is believed a promising source of renewable energy. At present, the development and applications of hydrogen energy have attracted great attention. The successful application of hydrogen energy at a large scale ultimately depends on efficient and sustainable production, storage and transport of dihydrogen [5,6,7,8]. Hydrogen storage materials are divided into chemical hydrogen storage materials and physical hydrogen storage materials. Among many chemical hydrogen storage materials, one of them is formic acid (HCOOH, FA), which has attracted considerable attention due to its economy, recyclability, low toxicity, low flammability, high stability and high hydrogen capacity (4.4 wt.%) under normal conditions [9,10]. For the decomposition reaction of FA, there are two possible pathways, dehydrogenation (Reaction (1)) and dehydration (Reaction (2)). It is obvious that the Reaction (1) is the only one desired if FA is used to produce H_2_. Reaction (2), on the contrary, is an off-putting one owing to the generation of CO, which is toxic and may deactivate catalysts and so on [11,12,13]. Therefore, the key to utilize FA as a hydrogen storage material is to develop low-cost and highly efficient catalysts for DFA.
(1)$$\mathrm{HCOOH}\to H_{2}+{\mathrm{CO}}_{2}$$
(2)$$\mathrm{HCOOH}\to H_{2}O+\mathrm{CO}$$

The United Nations’ sustainable development goals, including reducing carbon emissions, have provided a much-needed boost the development of catalysts with high selectivity and activity for DFA. At present, many catalysts have been reported for DFA [14,15,16,17,18], including homogeneous catalysts and supported mono-metallic, bi-metallic and trimetallic catalysts. Padhi and co-workers reported using a half-sandwich complex containing Ru to dehydrogenate formic acid. They believe that the complex is a descent catalyst towards dehydrogenation of formic acid into carbon dioxide and hydrogen in 1:1 ratio [19]. Beller et al., also reported catalytic dehydrogenation of FA with ruthenium–PNP–pincer complexes. They found the catalyst show highest activity in an acidic environment (pH 4.5) [20]. The cobalt pincer complex has also been reported for the dehydrogenation of formic acid in aqueous medium under mild conditions [21]. Many researchers reported heterogeneous catalysts for DFA. Amos and co-workers reported a non-transition-metal catalysis system for DFA, which exhibits very encouraging performance [22]. However, most of the catalysts reported for DFA are transition metal catalysts. Jiang et al., have reviewed heterogeneous catalysts used in DFA [23].

The current research relevant to heterogeneous catalysts for DFA mainly concerns the activities and lifetimes of catalysts, which is focused on active sites and supports. Au [24], Pd [25] and bimetallic combinations, such as PdAu and PdAg [26,27], are commonly used as catalytic active sites. Among them, Pd is regarded as a most promising active component. The widely used supports of catalysts for DFA are graphite, carbon, mesoporous-silica, TiO_2_, ZrO_2_, metal–organic frameworks (MOF) and so on [28,29,30,31,32,33,34]. These supports have large specific surface areas (SSAs), which is a benefit to dispersing metal particles, active sites, in catalysts, and prevents them from agglomerating effectively. As a support, g-C_3_N_4_ has also attracted much attention [35]. Though the SSA of g-C_3_N_4_ is not very large, the dπ–pπ interaction between g-C_3_N_4_ and transition metallic particles may change the electron density of catalytic active sites, which is beneficial for the formation of M-formate in the DFA.

Recently, a natural nanotube, halloysite [Al_2_Si_2_O_5_(OH)_4_·nH_2_O], which is an abundant and cheap natural resource in many countries [36,37,38], has attracted much attention due to its biocompatibility and large SSA. Halloysite nanotubes (HNTs) external surfaces are composed of siloxane (Si-O-Si) groups, whereas the internal surface consists of a gibbsite-like array of aluminol (Al-OH) groups [39]. Regrettably, this structure is not beneficial to the interaction between metal nanoparticles and HNTs, though HNTs are used as supports. However, it is the unique structure of HNTs that makes it easy to modify them with functional agents [40], which can greatly improve the adhesion of metal nanoparticles to their surfaces [41,42,43,44]. Various chemical reagents, such as sodium citrate, EDTA, 3-mercaptopropyltrimethoxysilane [45], 3-aminopropyltriethoxysilane (APTES) [46] and poly diallyldimethylammonium chloride [47,48,49], were used to modify HNTs in order to improve the performances of HNTs. Yuan et al., reported the modification of HNTs with γ-aminopropyltriethoxysilane (APTES) [50], by which the amino-functionalized HNTs were obtained via the condensation between APTES and the hydroxyl groups of HNTs, including Al–OH and Si–OH [51,52].

Based on the above-mentioned, in the present contribution we report the amino-functionalization of natural halloysite nanotubes and focus on the utilization of amino-functionalized HNTs as supports for supported metallic (Pd, Au, Ag) catalysts in the DFA. The -NH_2_ groups on the supports serve as basic sites, which facilitates not only the dispersion of metal nanoparticles, but also the formation of formate ions by absorbing H^+^ ions of formic acid [53], and then improving the interaction between the metal NPs and formate ions, enhancing the catalytic activity of catalysts in the DFA. 

## 2. Materials and Methods

### 2.1. Materials and Reagents

Aminopropyltriethyoxysilane (APTES, ≥98.0%) was purchased from NJDULY, Nanjing, China; sodium formate (HCOONa, 98.4%) was purchased from Beat Medicine, Shanghai, China. Halloysite nanotubes (Tianjin First Chemical Plant, Tianjin, China), sodium borohydride (NaBH_4_, ≥98.0% Tianjin First Chemical Plant), formic acid (HCOOH, FA, ≥98.0% Tianjin First Chemical Plant), sodium carbonate (Na_2_CO_3_, ≥99.8% Tianjin First Chemical Plant), toluene (C_7_H_8_, ≥99.5% Tianjin First Chemical Plant), ethanol (CH_3_CH_2_OH, ≥99.7% Tianjin Second Chemical Plant, Tianjin, China), ammonium tetrachloropalladate ((NH_4_)_2_PdCl_4_, 99.9%, Pd > 36.5%, Tianjin Heowns, Tianjin, China), silver nitrate (AgNO_3_, ≥99.8% Tianjin First Chemical Plant), chloroauric acid (HAuCl_4_, ≥47.8% Tianjin First Chemical Plant). The deionized water was obtained by reversed osmosis followed by ion-exchange and filtration.

### 2.2. Synthesis of Catalysts

#### 2.2.1. Amino-Functionalization of HNTs

The synthesis of amino-functionalized HNTs is described as follows. First of all, the natural HNTs were calcined at 110 °C for 12 h. Then, 1.2 g of calcined HNTs and 4 mL of APTES were added into 25 mL of anhydrous toluene under stirring at room temperature, followed by ultrasounding for 1 h; then, the mixture was heated at 80 °C for 20 h under stirring [54]. After centrifugation, the solid product was washed three times with toluene and dried at 80 °C. Product is labeled NH_2_-HNTs. 

#### 2.2.2. Synthesis of Pd/NH_2_-HNTs

First, 0.6 g of NH_2_-HNTs was dispersed in 4 mL of deionized water, into which 3 mL of (NH_4_)_2_PdCl_4_ (0.05 M) was added. The mixture was stirred for 12 h at room temperature after several vacuumizing and aerating cycle treatments under agitation. The solid collected by centrifugation was washed with water and ethanol, and finally dried at 80 °C, yielding Pd (II)-functionalized NH_2_-HNTs. The obtained material was then treated with 0.1g NaBH_4_ in 20 mL of Na_2_CO_3_ solution (0.025 M), forming NH_2_-HNT-supported Pd NPs [55]. The presence of Na_2_CO_3_ in solution makes the NaBH_4_ not easily hydrolyzed; a suitable pH value even facilitates the growth of uniform Pd nanoparticles. The product was labeled Pd/NH_2_-HNTs (Figure 1). The catalysts with different Pd contents were also prepared by changing the amounts of (NH_4_)_2_PdCl_4_. For comparison, the Pd/HNTs, Ag/NH_2_-HNTs and Au/NH_2_-HNTs were also prepared in the same preparation processes, except that the NH_2_-HNTs or Pd were replaced by the HNTs, Ag or Au, respectively.

#### 2.2.3. Synthesis of PdAu/NH_2_-HNTs

PdAu/NH_2_-HNTs were synthesized by co-impregnation method. First, 0.6 g of NH_2_-HNTs were dispersed in 4 mL of deionized water, into which 1.5 mL of (NH_4_)_2_PdCl_4_ (0.05 M) and 7.5 mL of HAuCl_4_ (0.01 M) were added. The synthesis processes followed were the same as those for preparing Pd/NH_2_-HNTs. The product obtained was labeled as PdAu/NH_2_-HNTs.

#### 2.2.4. Synthesis of PdAg/NH_2_-HNTs

First, 0.6 g of NH_2_-HNTs were dispersed in 4 mL of deionized water, into which 1.5 mL of (NH_4_)_2_PdCl_4_ (0.05 M) and 1.5 mL of AgNO_3_ (0.05 M) were added. The synthesis processes followed were the same as those of preparing Pd/NH_2_-HNTs. The product obtained was labeled as PdAg/NH_2_-HNTs.

### 2.3. Characterization

X-ray diffraction (XRD) patterns of samples were recorded on Rigaku SmartLab diffractometer (Rigaku Corporation, Tokyo, Japan) at room temperature, operating with Cu-Kα radiation, using a generator with a voltage of 40 KV and a current of 150 mA. Transmission electron microscopy (TEM) and energy dispersive spectroscopy (EDS) were performed on a Talos F200X G2 transmission electron microscope (Thermo Fisher Scientific Inc., Waltham, MA, USA) with an accelerating voltage of 200 kV, and the information resolution of the TEM was 0.12 nm. The BET surface area measurements were performed on Autosorb IQ C (Quantachrome Instruments, Boynton Beach, FL, USA) at 77 K. Prior to analysis, samples were degassed at 353 K for 4 h under vacuum. The H_2_ chemisorption was carried out on an Autosorb IQ C-XR. The measurement method was described in reference [35]. The mean size of metal particles in catalysts was calculated from the amount of H_2_ adsorbed. The Fourier transform infrared (FTIR) spectra were recorded on a Nicolet iS50 FT-IR (Thermo Fisher Scientific Inc., Waltham, MA, USA) in the wave number range from 1000 to 4000 cm^−1^. Inductively coupled plasma optical emission spectroscopy (ICP-OES) measurements were performed on Thermo Fisher iCAP 7000 (Thermo Fisher Scientific Inc., Waltham, MA, USA). X-ray photoelectron spectroscopy (XPS) spectra were acquired with Axis Ultra DLD (Shimadzu, Kyoto, Japan) using Al Kα X-rays, and the binding energy was calibrated by taking the C1s peak at 284.8 eV as the reference. The gas products were qualitatively analyzed by gas chromatography (GC) on a GC-7900 (Techcomp Instrument Limited, Shanghai, China) with a thermal conductivity detector (TCD).

### 2.4. Evaluation of Catalytic Activity of Catalysts in DFA 

Figure 1 shows the device for evaluating the catalytic performance of catalysts in the DFA. The experimental processes are described briefly as follows. First, 0.1 g of as-prepared catalyst was added into the round-bottom flask with three necks. Then, 20 mL of an aqueous formic acid/sodium formate (FA/SF) solution (C_FA_ + C_SF_ = 6.0 M) was added into the pressure-equalization funnel. The gas burette was filled with NaOH solution in order to absorb the CO_2_ produced in the reaction. The catalytic reaction started as soon as the FA/SF solution was added into the flask under stirring. The evolution of gas was monitored by gas burette. The catalytic reactions were carried out at different temperatures (283, 298, 313 K) by adjusting the temperature of the water bath. The hydrostatic pressure formed due to the presence of the solution in the burette; the vapor pressure of water was deducted; and the total pressure of the H_2_ collected was assumed constant throughout the experiment at atmospheric pressure. All calculations were performed based on the ideal gas law [56].

In order to study the durability and recyclability of the catalyst, the catalyst was collected after the initial catalytic test, washed with deionized water and then dried at room temperature for 6 h under vacuum. The recovered catalyst was then used in the next reaction under the same conditions as above.

### 2.5. Turn-Over-Frequency (TOF) Calculations

The TOF of the catalyst was calculated based on the following formula.
TOFpd = (P V/R T)/n t
where TOF_Pd_ is the TOF (n_H2_/n_Pd_/h) at time t, P is the atmospheric pressure (Pa), V is the volume of the generated H_2_ (m^3^), R is the universal gas constant (8.3145 m^3^·Pa·mol^−1^·K^−1^), T is the temperature of gas (K), n is the molar quantity of Pd in the catalyst and t is reaction time (h).

## 3. Results

The SSAs and pore volumes (Vp) of HNTs, NH_2_-HNTs, Pd/HNTs, Pd/NH_2_-HNTs, PdAu/NH_2_-HNTs and PdAg/NH_2_-HNTs are listed in Table 1. The adsorption isotherms of HNTs, NH_2_-HNTs and Pd/NH_2_-HNTs are type Ⅱ with H3 hysteresis loops according to the IUPAC-classification (Appendix A). Compared with the SSA (62.3 m^2^·g^−1^) and Vp (0.41 cm^3^·g^−1^) of the original HNTs, those (55.9 m^2^·g^−1^, 0.40 cm^3^·g^−1^) of Pd/HNTs are low, for which the large Pd particles blocking tubes or entering into the inner parts of the tubes should be responsible. Similarly, the SSA (34.8 m^2^·g^−1^) of NH_2_-HNTs is also smaller, which indicates that APTES was attached to the inner and outer surfaces of HNTs, reducing the SSA of HNTs. However, the SSA of NH_2_-HNT-loaded metal particles, e.g., Pd/NH_2_-HNTs, is larger than that of NH_2_-HNTs. This can be explained by the fact that the metal particles supported on the NH_2_-HNTs are very small and have high dispersion due to the presence of -NH_2_ groups, which do not block tubes but increase the SSA of the catalyst. Thus, the SSA should be sum of the original SSA of NH_2_-HNTs and the SSA of small metal particles [57]. It can be seen in the data in Table 1 that there are differences in the Vp of the catalysts, but the differences are inconspicuous. The mean particle sizes of Pd NPs in the as-prepared Pd/HNTs and Pd/NH_2_-HNTs determined by static chemisorption method were 3.5 and 1.2 nm respectively. 

In order to detect the structural changes of halloysite in all samples, XRD was used to analyze samples. Figure 2a depicts the XRD patterns of HNTs, NH_2_-HNTs, Pd/NH_2_-HNTs, PdAu/NH_2_-HNTs and PdAg/NH_2_-HNTs.

As shown in Figure 2a, we found primary halloysite displays peaks at 12.1°, 20.1°, 24.6°, 35.0°, 54.5° and 62.6°, which are consistent with the data of PDF#29-1487. It can be seen clearly that all peaks in NH_2_-HNTs correspond to those of HNTs, which indicates that NH_2_-HNTs maintain the phase structure of HNTs. That is to say, the reaction between APTES and HNTs does not damage the skeleton structure of HNTs; the reaction should take place on the surfaces of HNTs. All catalysts (Pd/NH_2_-HNTs, PdAu/NH_2_-HNTs, PdAg/NH_2_-HNTs) showed no other peaks, e.g., peaks of Pd, besides those of HNTs, which means that the metallic particles in catalysts were small or in a state of high dispersion.

The XRD of NH_2_-HNTs confirms that the skeleton structure of HNTs was maintained in NH_2_-HNTs; however, information about amino-functional groups in NH_2_-HNTs is absent. In order to obtain the information about amino-functional groups in samples and explore the bonding between APTES and HNTs, FTIR was used to characterize samples. Figure 2b shows the FTIR spectra of HNTs, NH_2_-HNTs and Pd/NH_2_-HNTs. The frequencies and assignments of each vibrational mode observed are listed in Appendix A.

Figure 2b displays the typical vibration bands of halloysite, which are in agreement with those reported by reference [54]. Compared with HNTs, NH_2_-HNTs exhibited several new FTIR peaks, such as the stretching vibration and the deformation vibration bands of CH_2_ at round 2930 and 1490 cm^−1^, respectively, and a peak at 1570 cm^−1^ that is attributed to the deformation vibration of N–H bonds. All of these peaks indicate the presence of moieties of APTES in NH_2_-HNTs, the amino-functionalized HNTs. Furthermore, the O–H stretching band (3626 cm^−1^) in HNTs is highlighted. It is the -OH groups on the surfaces of HNTs that make the reaction between HNTs and APTES possible [50].

TEM images and the element distribution mappings of the catalysts are shown in Figure 3 and Appendix A. The natural HNTs were mainly composed of hollow nanotubes ranging from 0.5 to 1.0 μm in length, with internal diameters of 10–30 nm. Figure 3a clearly shows that NH_2_-HNTs, the HNTs modified by APTES, still maintained a perfect tubular structure, which confirms that the reaction between APTES and HNTs does not damage the tubular structure of the HNTs. It can be seen clearly in Figure 3b,c (Pd/NH_2_-HNTs) that some black particles were uniformly deposited and dispersed on the surfaces of the tubes, which should be Pd particles. Their size as evaluated using ImageJ, was about 1.8 ± 0.4 nm, which is similar to the result obtained by static chemisorption (1.2 nm). However, the Pd NPs in Pd/HNTs prepared under the same synthetic conditions as those of Pd/NH_2_-HNTs were all larger than 4.3 nm, and the metal particles were obviously aggregated (Figure 3d), even though the Pd content characterized by ICP in Pd/HNTs (0.60 wt.%) was lower than that in Pd/NH_2_-HNTs (2.50 wt.%). These results indicate that the H_2_N-groups on the surfaces of HNTs have significant influences on the size and dispersion of Pd particles, and Pd loading amount. The interaction between PdCl_4_^2−^ and NH_2_-HNTs is different from that between PdCl_4_^2−^ and HNTs because of the difference in surface structure between HNTs and NH_2_-HNTs, which causes in aforementioned results. For NH_2_-HNTs, the -CH_2_CH_2_CH_2_NH_2_ connected to the surfaces of HNTs are readily transformed into -CH_2_CH_2_CH_2_NH_3_^+^ [58], which has strong electrostatic attraction to PdCl_4_^2−^.

Furthermore, the -NH_2_ groups, Lewis bases, can promote the interactions among the metal Pd particles, Lewis acid and support, making a contribution to the high dispersion and loading of ultrafine particles. However, the small Pd particles in Pd/HNTs tended to aggregate and can form larger particles; the maximum size of particles was ca. 25 nm. Figure 3e,f shows the distribution mappings of Pd element in Pd/NH_2_-HNTs, which further demonstrate the existence and high dispersion of Pd in the catalysts. The HAADF-STEM images and the corresponding element mapping of PdAu/NH_2_-HNTs are shown in Figure 3g–i. As shown in these images, Pd and Au were uniformly dispersed in the NH_2_-HNTs. It can be seen in the distribution mappings of Pd and Ag elements in PdAg/NH_2_-HNTs (Appendix A) that the metallic particles were successfully and uniformly anchored on the modified HNTs.

To further investigate the presence of -NH_2_ groups, the chemical states of metallic elements, and the interactions between metals and the supports in the prepared catalysts, XPS was also used to characterize samples. As shown in Figure 4a, in addition to peaks of C, O, Al and Si, an obvious peak of N (BE∼400 eV) was observed in Pd/NH_2_-HNT and NH_2_-HNT spectra, but this N peak did not exist in the HNTs’ spectrum.

Figure 4b–e displays the XPS spectra of Pd 3d in samples. The XPS spectra of Ag and Au in samples are shown in Appendix A.

The peaks centered at 335.2 and 337.2 eV are attributed to the 3d_5/2_ of Pd^0^ and 3d_5/2_ of Pd^2+^, respectively (Figure 4b) [59]. Pd exists as Pd^0^ and Pd^2+^ in the Pd/HNTs, but the Pd^0^ is dominant based on the peak area ratios of the peaks. Compared with Figure 4b, Figure 4c shows that all peaks attributed to Pd^0^ shifted toward higher binding energies for Pd/NH_2_-HNTs, for which a reasonable explanation is that the Pd particles are smaller and attached to -NH_2_ groups in Pd/NH_2_-HNTs [60]. The electronegativity of N is higher than that of Pd, so partial electron migration from Pd to N may take place while there is an interaction between them, which would also help Pd to interact better with negatively charged intermediates, such as formate ions, in the DFA. It is worth noting that the ratio of nPd^2+^ to nPd^0^ in Pd/NH_2_-HNTs is higher than that in Pd/HNTs (Table 2).

These results are reasonable because the Pd particles in Pd/NH_2_-HNTs are smaller than those in Pd/HNTs (cf. TEM). Smaller Pd particles that have larger SSAs are more unstable and readily oxidized by oxygen. However, the PdO species are readily reduced in hydrogen atmosphere even at temperatures below 30 °C [61], so a small amount of PdO in a catalyst has no effect on its catalytic activity in FAD. Similarly, Pd 3d peaks in Figure 4d,e are also shifted. However, it can be seen that the influences of Ag and Au on the shift of Pd 3d peaks are different in comparison with Figure 4b,c, which can be explained by the difference in work functions of the metals [62]. The work functions of Ag, Pd and Au are 4.26, 5.12 and 5.10 eV, respectively. It is well known that the smaller the work function of the metal, the easier the metal loses electron. When two metals with different work functions combine together, the electrons may migrate from the metal with the lower work function to the one with the higher work function. Accordingly, the binding energies of Pd 3d_5/2_ and 3d_3/2_ in Pd/NH_2_-HNTs (335.8 and 341.0 eV) are higher and lower than those in PdAg/NH_2_-HNTs (335.7 and 341.0 eV) and PdAu/NH_2_-HNTs (336.1 and 341.3 eV), respectively. All XPS results strongly confirm the existence of strong interactions between metals and metal and support. The electronic effect between Pd and Au and the interaction between metal and support may produce a synergistic effect and significantly enhance the catalytic performance of PdAu/NH_2_-HNTs for DFA.

The catalytic performances of all catalysts for DFA were evaluated. Table 3 and Figure 5 show the catalytic performances of as-prepared catalysts at 298 K. It should be noted that no CO was detected by GC in any of the collected gas (Appendix A), which means the FA decomposes into H_2_ and CO_2_ over the catalysts with high selectivity.

Obviously, all catalysts containing Pd showed catalytic activity (Entries 1–6), whereas those without Pd did not show catalytic activity (Entries 7 and 8). Pd was undoubtedly the catalytic active center for DFA. In order to explore the influences of -NH_2_ groups, catalysts with similar Pd contents were designed and compared. The results display that the catalytic activity of Pd (1.28%)/HNTs (Entry 2) is obviously weaker than that of Pd (1.30%)/NH_2_-HNTs (Entry 4), though they have similar Pd contents. This is due to the presence of -NH_2_ groups serving as basic sites. The order of the activity of three catalysts with the same Pd content and support (Entries 4, 5 and 6) was as follows: PdAu/NH_2_-HNTs > Pd/NH_2_-HNTs > PdAg/NH_2_-HNTs, which is agreement with the order of binding energies of Pd^0^ in these catalysts (cf. XPS). It confirms that the formate ions are firstly combined with metal Pd as intermediates. In this case, reducing the electron density on the surfaces of Pd NPs is beneficial to the combining of catalytic active centers and formate ions, and then enhances the catalytic activity of all the catalysts. Accordingly, PdAu/NH_2_-HNTs showed the highest activity among all the catalysts. In addition, the higher the Pd contents in catalysts with the same support (Entries 1 and 2; Entries 3 and 4), the larger the volume of H_2_ evolved in the DFA in the same reaction time (70 min).

The above results suggest that the -NH_2_ in Pd/NH_2_-HNTs plays an important role in enhancing the catalytic activity of Pd/NH_2_-HNTs for DFA. This excellent catalytic performance can be ascribed to three factors. Firstly, the -NH_2_ groups are conducive to the combination and connection between the support and PdCl_4_^2−^ ions, which promotes the fixation and dispersion of Pd, leading to ultrafine Pd particles in the catalyst. Secondly, the alkalinity of -NH_2_ groups is favorable for the breaking of the O-H bonds of FA, forming formate ions, which improves the catalytic process. Thirdly, the synergistic interactions among Pd active centers, -NH_2_ groups on the support and metal of higher work function may change the electron density of Pd^0^, which is beneficial to the combining of formate ions and catalyst. In summary, the -NH_2_ groups on the support are able to enhance the catalytic activity of the catalyst.

To explore the activity of the catalyst at different temperatures, the activities of the catalysts for DFA were evaluated and compared at 283, 298 and 313 K. Taking Pd/NH_2_-HNTs as an example, its activities at different temperature are shown in Figure 5b, which displays clearly that the rate of hydrogen generation increased as reaction temperature increased. What is interesting is that the catalyst did show excellent catalytic activity at 283 K, producing H_2_ 112 mL, which means that this catalyst has some potential practical value for DFA breakdown. The molar ratio of FA to SF has significant influence on the performance of Pd/NH_2_-HNTs in the DFA (Figure 5c). In the FA-SF aqueous solution, the activity of the catalyst for DFA increased with the molar fraction of SF, and 50% was the optimal value. This demonstrates that the SF serves as a catalyst promoter to accelerate the reaction rate. Firstly, the formate ions in the reaction system coordinate with active sites, forming the key intermediate, M-formate; and then, accompanied by *β*-H transfer, it is turned into hydride (M-H) and CO_2_. The formed hydride reacts with H^+^ ions in the reaction system immediately, releasing H_2_.

The recyclability of catalyst was investigated under optimal reaction conditions. The results are shown in Figure 5d.

As shown in Figure 5d, the catalytic activity of catalyst decreased slightly after the first run. However, the activity was still stable after the second run. The TEM image of the Pd/NH_2_-HNTs after the fifth run shows that the Pd NPs were still in good dispersion on the support, though the size of Pd NPs increased (Appendix A). Further work on enhancement of the stability of the present catalyst for the DFA is underway.

## 4. Conclusions

In summary, we have developed a new catalyst that uses a cheap natural alloy site as a support to well disperse Pd nanoparticles. It is the hydroxyl groups on HNTs that make the connection between APTES and HNTs possible, so Pd nanoparticles can be evenly dispersed on the NH_2_-functionalized HNTs. The obtained catalysts, Pd/NH_2_-HNTs, PdAu/NH_2_-HNTs and PdAg/NH_2_-HNTs, showed excellent catalytic performance in DFA, which demonstrates that the -NH_2_ group may change the electron density of Pd and facilitate the decomposition of HCOOH. This study showed that the size of Pd particles in supported catalysts is strongly dependent on the support. It is expected that combining the precious metal Pd with other non-noble metals with higher work functions may reduce the cost of catalysts for DFA.

## Data Availability

The data presented in this study are avilable upon request from the corresponding author.

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
