# Peer review of "Amine-Functionalized Natural Halloysite Nanotubes Supported Metallic (Pd, Au, Ag) Nanoparticles and Their Catalytic Performance for Dehydrogenation of Formic Acid"

_nanomaterials, 2022, doi:10.3390/nano12142414_

Round 1

Reviewer 1 Report

In recent years, the production of hydrogen by dehydrogenation of formic acid (FA) has been a very popular topic. Palladium containing catalysts are the most active and selective in this reaction. The work proposed for publication is one of such works. The Pd catalysts developed by the authors make it possible to produce hydrogen by the decomposition of FA at low temperatures. This allows the authors to state that the catalysts have some potential practical value in dehydrogenation of FA.  And one could agree with this if it were not for the deactivation of the resulting catalysts, about which the authors do not write anything. If we look at the results of the catalytic experiments presented in Figure 5, we can see that after the release of 160-170 ml of hydrogen on the best catalysts, the reaction rate drops sharply. This amount of hydrogen released corresponds to 10-12% conversion of formic acid, after which the catalyst stops working. The authors should pay attention to this fact and try to explain the causes of deactivation and the method of catalyst reactivation.

Author Response

Thank you for your comment! Our catalysts do need further improvement in terms of catalytic life. We do have investigated the cause of catalyst deactivation in the recycling. There are two main causes that result in catalyst deactivation in the recycling. The first is that the Pd particles, catalytic active centers, are easily detached from the support during the catalytic reaction. The second is that the size of metal particle increases, which leads to the decrease of its catalytic activity in the reaction.

The work improving the catalytic activity of the catalyst in the recycling is still being further explored.

Reviewer 2 Report

B. Zhu, W. Huang and co-authors reported the preparation of a new catalysts based on natural halloysite as support and disperse Pd (as well as PdAu and PdAg) nanoparticles. NH2-functionalization of halloysite was applied to obtain a fine distribution of metal nanoparticles. The obtained catalysts were tested in the formic acid dehydrogenation reaction and showed a high selectivity of the reaction proceeding without CO evolution, while the activity of the catalysts varied in the series PdAu/NH2-HNTs > Pd/NH2-HNTs > PdAg/NH2-HNTs. The authors comprehensively studied the composition and structure of the obtained materials and studied in detail the catalytic activity, including the temperature dependence and the possibility of catalyst recycling. The work is fully consistent with the profile of the journal and can be published in Nanomaterials.

Small comments:

Page 3, fix title, should be 2.2.3 Synthesis of PdAu/NH2-HNTs

Page 4, line 149: described

Page 4, line 161: the abbreviation SF is firstly appeared and should be explained

Page 7-8: the authors explain the greater dispersity of the distribution of Pd nanoparticles for Pd/NH2-HNTs catalyst compared to Pd-HNTs by the electrostatic interaction of -NH3+ groups with [PtCl4]2- during the formation of the material precursor. This is probably also true for the use of [AuCl4]-. At the same time, the Ag+ cationic precursor was used to precipitate Ag - did this somehow affect the dispersity and distribution of metal nanoparticles in the PdAg/NH2-HNTs material?

Author Response

Answer:

Thank you for your comment! Your comments are very helpful to improve the level of the manuscript! We have revised the manuscript according to your comments. We have also considered this problem before, and prepared PdAg catalyst, PdAg/NH2-HNTs, with small particles and large dispersion by adjusting the way of preparation, as shown in Figure S2. We can see that the metal particles in the prepared catalyst are really very small, which is agreement with the references. It is believed that the catalytic activity of the catalyst will not be affected by this preparing method.

Reviewer 3 Report

1. Highlight the importance of reactions between the HNTs and APTES with presence of OH groups.

2. Authors needs to explicitly mention the reasoning behind the functionalization of Amine groups and interaction between the metal particles.

3. Authors need to discuss more in detail with equations  about electronic effects between Pd/Au  with HNTs and the interactions that is causing the synergistic effect and schematic sketch of the electron migrations wrt to work functions.

4. What causes drop in volume with PdAg  over PdAu/NH3-HNTs in Fig. 5a

5. How did the authors optimize the wt. % of Pd ?

6. Above all these Fig.5 tests results should be compared with previous available  literatures.

Author Response

  1. Highlight the importance of reactions between the HNTs and APTES with presence of OH groups.

Answer:

It can be found by comparing functions of HNTs modified and not modified with -NH2 that the modification with organic silane plays a crucial role in the improvement of catalyst activity. According to references [[50, 54], the presence of –OH groups can help organosilane attach on the surface of HNTs by the Si-O bonds, which makes APTES modification on the halloysite nanotubes.

  1. Authors needs to explicitly mention the reasoning behind the functionalization of Amine groups and interaction between the metal particles.

Answer:

Lines 263-265 describe the interaction between -NH2 groups and PdCl42- ion, which is benefit to the increase in the loading amount of Pd and high dispersion of Pd on the support. Lines 283-292 and 350-364 mention the interaction between -NH2 groups and metal Pd to improve the catalytic activity of catalysts.

  1. Authors need to discuss more in detail with equations about electronic effects between Pd/Au with HNTs and the interactions that is causing the synergistic effect and schematic sketch of the electron migrations wrt to work functions.

Answer:

Thank you for your comment! Your comment is very helpful to improve the level of manuscript. For the electronic effect among Pd, Au and HNTs, it is a complex system not easy to calculate by a simple equation. However, the literature and experimental data in XPS confirm the presence of electronic effect between Pd/Au and HNTs. In the next work, theoretical calculation will be carried out to discuss the influence of electronic effect among Pd, Au and HNTs in detail.

  1. What causes drop in volume with PdAg over PdAu/NH3-HNTs in Fig. 5a

Answer:

As mentioned in lines 324-339, the electron density of metal Pd in PdAg is slightly higher than that in PdAu, which is not conducive to the combination between formate ion and metal Pd, resulting in the drop in volume with PdAg over PdAu/NH2-HNTs in Fig. 5a.

  1. How did the authors optimize the wt. % of Pd?

Answer:

We optimize the wt. % of Pd by two ways, previous work of our group and literature data. In this paper, we focus on the influence of support modification and electronic effect between bimetals on catalytic activity of catalysts, so the influence of metal content in catalysts is not emphatically discuss.

  1. Above all these Fig.5 tests results should be compared with previous available literatures.

Answer:

Relevant content has been added in the Supporting Information according to your comments.

Round 2

Reviewer 1 Report

I am satisfied with the answers of the authors. The manuscript may be accepted in present form.